# MODEL ENSEMBLE-BASED INTRINSIC REWARD FOR SPARSE REWARD REINFORCEMENT LEARNING

## ABSTRACT

In this paper, a new intrinsic reward generation method for sparse-reward reinforcement learning is proposed based on an ensemble of dynamics models. In the proposed method, the mixture of multiple dynamics models is used to approximate the true unknown transition probability, and the intrinsic reward is designed as the minimum of the surprise seen from each dynamics model to the mixture of the dynamics models. In order to show the effectiveness of the proposed intrinsic reward generation method, a working algorithm is constructed by combining the proposed intrinsic reward generation method with the proximal policy optimization (PPO) algorithm. Numerical results show that for representative locomotion tasks, the proposed model-ensemble-based intrinsic reward generation method outperforms the previous methods based on a single dynamics model.

## 1    INTRODUCTION

Reinforcement learning (RL) with sparse reward is an active research area (Andrychowicz et al., 2017; de Abril & Kanai, 2018; Kim et al., 2018; Oh et al., 2018; Tang et al., 2017). In typical model-free RL, an agent learns a policy to maximize the expected cumulative reward under the circumstance that the agent receives a non-zero reward from the environment for each action of the agent. On the contrary, in sparse reward RL, the environment does not return a non-zero reward for every action of the agent but returns a non-zero reward only when certain conditions are met. Such situations are encountered in many action control problems (Andrychowicz et al., 2017; Houthooft et al., 2016; Oh et al., 2018). As in conventional RL, exploration is important at the early stage of learning in sparse reward RL, whereas the balance between exploration and exploitation is required on the later stage. Methods such as the $\epsilon$-greedy strategy (Mnih et al., 2015; Van Hasselt et al., 2016) and the control of policy gradient with Gaussian random noise (Duan et al., 2016; Schulman et al., 2015a) have been applied to various tasks for exploration. However, these methods have been revealed to be insufficient for successful learning when reward is sparse (Achiam & Sastry, 2017).

In order to overcome such difficulty, intrinsically motivated RL has been studied to stimulate better exploration by generating intrinsic reward for each action by the agent itself, even when reward is sparse. Recently, many intrinsically-motivated RL algorithms have been devised to deal with the sparsity of reward, e.g., based on the notion of curiosity (Houthooft et al., 2016; Pathak et al., 2017) and surprise (Achiam & Sastry, 2017). It is shown that these algorithms are successful and outperform the previous approaches. In essence, these algorithms use a single estimation model for the next state or the environment dynamics to generate intrinsic reward.

In this paper, in order to further improve the performance of sparse reward model-free RL, we propose a new method to generate intrinsic reward based on *an ensemble of estimation models for the environment dynamics*. The rationale behind our approach is that by using a mixture of several distributions, we can increase degrees of freedom for modeling the unknown underlying model dynamics and designing a better reward from the ensemble of estimation models. Numerical results show that the proposed model-ensemble-based intrinsic reward generation method yields improved performance as compared to existing reward generation methods for continuous control with sparse reward setting.

## 2 THE PROPOSED METHOD

### 2.1 PRELIMINARIES

In this paper, we consider a discrete-time continuous-state Markov Decision Process (MDP), denoted as $(\mathcal{S}, \mathcal{A}, P, r, \rho_0, \gamma)$, where $\mathcal{S}$ and $\mathcal{A}$ are the sets of states and actions, respectively, $P : \mathcal{S} \times \mathcal{A} \times \mathcal{S} \to [0, 1]$ is the transition probability function (called model dynamics), $r : \mathcal{S} \times \mathcal{A} \times \mathcal{S} \to \mathbb{R}$ is the *extrinsic* reward function, $\rho_0 : \mathcal{S} \to [0, 1]$ is the distribution of the initial state, and $\gamma$ is the discounting factor. A (stochastic) policy is represented by $\pi : \mathcal{S} \times \mathcal{A} \to [0, 1]$, where $\pi(a|s)$ represents the probability of choosing action $a \in \mathcal{A}$ for given state $s \in \mathcal{S}$. In sparse reward RL, the environment does not return a non-zero reward for every action but returns a non-zero reward only when certain conditions are met by the current state, the action and the next state (Andrychowicz et al., 2017; Houthooft et al., 2016; Oh et al., 2018). The goal of this paper is to optimize the policy $\pi$ to maximize the expected cumulative return $\eta(\pi)$ by properly generating intrinsic reward in such sparse reward environments. We assume that the true transition model $P$ is unknown to the agent.

Intrinsically-motivated RL adds a properly designed intrinsic reward to the actual extrinsic reward to yield a non-zero total reward for training even when the extrinsic reward returned by the environment is zero (de Abril & Kanai, 2018; Pathak et al., 2017; Tang et al., 2017). One way to design such an intrinsic reward for action control is based on *surprise*, which is a measure of the unexpectedness of observing the next state for a given current state and action pair and is especially useful to yield better exploration (Achiam & Sastry, 2017). In this context, a recent work (Achiam & Sastry, 2017) proposed a promising direction of intrinsic reward design in which the agent tries to optimize its policy $\pi$ according to

$$\max_{\pi} \left\{ \eta(\pi) + c \, \mathbb{E}_{(s,a) \sim \pi}[D_{KL}(P||P_\phi)|(s,a)] \right\} \tag{1}$$

for some constant $c > 0$, where $P_\phi$ is the learning model parameterized by $\phi$ that the agent has regarding the true unknown transition probability $P$ of the environment. $D_{KL}(P||P_\phi)|(s,a)$ is the Kullback-Leibler divergence (KLD) between two distributions $P$ and $P_\phi$ of the next state for given current state-action pair $(s,a)$, and $\mathbb{E}_{(s,a) \sim \pi}$ is the expectation over $(s,a)$ following the policy $\pi$. Thus, the surprise is quantified as $\mathbb{E}_{(s,a) \sim \pi}[D_{KL}(P||P_\phi)|(s,a)]$ (Achiam & Sastry, 2017). Furthermore, the KLD $D_{KL}(P||P_\phi)|(s_t, a_t)$ at timestep $t$ can be lower-bounded as

$$D_{KL}(P||P_\phi)|(s_t, a_t) \geq \mathbb{E}\left[\log \frac{P_{\phi'}(\cdot|s_t, a_t)}{P_\phi(\cdot|s_t, a_t)}\right] \tag{2}$$

with an arbitrary choice of the parameter $\phi'$. Therefore, the intrinsic reward at timestep $t$ is determined as $r'_{t,int}(s_t, a_t, s_{t+1}) = \log \frac{P_{\phi'}(s_{t+1}|s_t, a_t)}{P_\phi(s_{t+1}|s_t, a_t)}$ (Achiam & Sastry, 2017), where $P_{\phi'}$ needs to be designed properly. With $\phi' = \phi(t)$ and $\phi = \phi(t^-)$, where $P_{\phi(t)}$ and $P_{\phi(t^-)}$ are respectively the agent's model for $P$ at timestep $t$ and the model before the update at timestep $t$, the intrinsic reward is given by the computable quantity named as the 1-step surprise:

$$r_{t,int}^{1-step}(s_t, a_t, s_{t+1}) = \log \frac{P_{\phi(t)}(s_{t+1}|s_t, a_t)}{P_{\phi(t^-)}(s_{t+1}|s_t, a_t)}. \tag{3}$$

The proposed 1-step intrinsic reward performs well compared to the previously designed intrinsic reward, and it is based on a single model $P_\phi$ for $P$, where $P_\phi$ for given $(s,a)$ is modeled as Gaussian distribution (Achiam & Sastry, 2017).

### 2.2 INTRINSIC REWARD DESIGN FROM AN ENSEMBLE OF DYNAMICS MODELS

In this paper, we take the principle that $D_{KL}(P||P_\phi)|(s,a)$ is a reasonable measure for surprise to promote exploration, and generalize the intrinsic reward design under this measure. However, instead of using a single learning model for $P$ as in the previous approach, we propose using *an ensemble of K dynamics models* $P_{\phi^1}, \cdots, P_{\phi^K}$ for $P$, constructing the mixture distribution

$$P_K = \sum_{i=1}^{K} q_i P_{\phi^i} \tag{4}$$

with the mixing coefficients $q_i \geq 0$ and $\sum_{i=1}^K q_i = 1$, and using $P_K$ in (4) as an estimate for the true unknown $P$. The rationale behind this is that by using a mixture of several distributions we increase degrees of freedom for modeling the underlying model dynamics and designing a better intrinsic reward. For the $j$-th model $P_{\phi^j}$, $j = 1, \cdots, K$, we have

$$D_{KL}(P||P_{\phi^j})|(s_t, a_t) \geq \mathbb{E}\left[\log \frac{P_K(\cdot|s_t, a_t)}{P_{\phi^j}(\cdot|s_t, a_t)}\right] \tag{5}$$

as in (2). Thus, for $P_{\phi^j}$, the intrinsic reward at timestep $t$ is determined as

$$\tilde{r}_{t,int}^j(s_t, a_t, s_{t+1}) = \log \frac{P_K(s_{t+1}|s_t, a_t)}{P_{\phi^j}(s_{t+1}|s_t, a_t)}. \tag{6}$$

Furthermore, (6) can be modified to yield a 1-step surprise intrinsic reward as

$$r_{t,int}^j(s_t, a_t, s_{t+1}) = \log \frac{\sum_{i=1}^K q_i P_{\phi_{l(t)}^i}(s_{t+1}|s_t, a_t)}{P_{\phi_{l(t)-1}^j}(s_{t+1}|s_t, a_t)} \tag{7}$$

where $P_{\phi_{l(t)}^j}$ and $P_{\phi_{l(t)-1}^j}$ are the $j$-th model at the update period $l$ corresponding to timestep $t$ and the previous update period $l-1$, respectively ($l(t)$ will become clear in the subsection 2.3).

Since the mixture model (4) has the increased model order for modeling the underlying dynamics distribution beyond single-mode distributions, we have more freedom to design intrinsic reward. That is, we now have $K$ values, $r_{t,int}^j(s_t, a_t, s_{t+1})$, $j = 1, \cdots, K$, for candidates for intrinsic reward. In order to devise a proper use of this extra freedom, we consider the following two objective functions:

$$\eta(\pi) = \mathbb{E}_{\tau \sim \pi}\left[\sum_t \gamma^t r(s_t, a_t, s_{t+1})\right] \tag{8}$$

$$\tilde{\eta}(\pi) = \mathbb{E}_{\tau \sim \pi}\left[\sum_t \gamma^t r(s_t, a_t, s_{t+1})\right] + c\,\mathbb{E}_{(s,a)\sim\pi}\left[D_{KL}\left(P||P'\right)|(s,a)\right] \tag{9}$$

where $\tau$ is a sample trajectory, $c$ is a positive constant, and $P(\cdot|s, a)$ and $P'(\cdot|s, a)$ are the true transition probability of an environment and its estimation model, respectively. The first objective function $\eta(\pi)$ is the actual desired expected cumulative return for policy $\pi$ and the second objective function $\tilde{\eta}(\pi)$ is the expected cumulative sum of the actual reward and intentionally-added surprise for policy $\pi$. We define $\pi^*$ and $\tilde{\pi}^*$ as optimal solutions which maximize the objective functions (8) and (9), respectively. Note that with additional intrinsic reward, the agent learns $\tilde{\pi}^*$. Regarding $\eta(\pi^*)$ and $\eta(\tilde{\pi}^*)$, we have the following proposition:

**Proposition 1.** *Let $\eta(\pi)$ be the actual expected discounted sum of extrinsic rewards defined in (8). Then, the following inequality holds:*

$$0 \leq \eta(\pi^*) - \eta(\tilde{\pi}^*) \leq c\,\mathbb{E}_{(s,a)\sim\tilde{\pi}^*}\left[D_{KL}\left(P||P'\right)|(s,a)\right] \tag{10}$$

*where $c$ is a positive constant.*

Proposition 1 implies that better estimation of the true transition probability $P$ by model $P'$ makes $\eta(\tilde{\pi}^*)$ closer to $\eta(\pi^*)$, where $\tilde{\pi}^*$ is learned based on $\tilde{\eta}(\pi)$. Thus, for given $P$ we want to minimize $\mathbb{E}_{(s,a)\sim\tilde{\pi}^*}\left[D_{KL}\left(P||P'\right)|(s,a)\right]$ in (10) over our estimation model $P'$ so that we have a tighter gap between $\eta(\pi^*)$ and $\eta(\tilde{\pi}^*)$, and the policy $\tilde{\pi}^*$ learned with the aid of surprise intrinsic reward well approximates the true optimal policy $\pi^*$. Regarding this minimization for tight gap, we have the following proposition:

**Proposition 2.** *Let $P_{\phi^i}(\cdot|s, a)$, $i = 1, \ldots, K$ be the ensemble of model distributions, and $P(\cdot|s, a)$ be an arbitrary true transition probability distribution. Then, the minimum of average KLD between $P(\cdot|s, a)$ and the mixture model $P' = \sum_i q_i P_{\phi^i}(\cdot|s, a)$ over the mixture weights $\{q_1, \cdots, q_K | q_i \geq 0, \sum_i q_i = 1\}$ is upper bounded by the minimum of average KLD between $P$ and $P_{\phi^i}$ over $\{i\}$: i.e.,*

$$\min_{q_1,\cdots,q_K} \mathbb{E}_{(s,a)\sim\tilde{\pi}^*}\left[D_{KL}\left(P \,\Big\|\, \sum_i q_i P_{\phi^i}\right)\Big|(s,a)\right] \leq \min_i \mathbb{E}_{(s,a)\sim\tilde{\pi}^*}\left[D_{KL}\left(P||P_{\phi^i}\right)|(s,a)\right]. \tag{11}$$

As seen in the proof of Proposition 2 in Appendix A, $\min_i \mathbb{E}_{(s,a)\sim\widetilde{\pi}^*}\left[D_{KL}\left(P||P_{\phi^i}\right)|(s,a)\right]$ provides the tightest upper bound on $\min_{q_1,\cdots,q_K} \mathbb{E}_{(s,a)\sim\widetilde{\pi}^*}[D_{KL}(P \parallel \sum_i q_i P_{\phi^i})|(s,a)]$ within the class of linear combinations of the individual surprise values $\{\mathbb{E}_{(s,a)\sim\widetilde{\pi}^*}\left[D_{KL}\left(P||P_{\phi^i}\right)|(s,a)\right],\ i = 1, 2, \cdots, K\}$. Propositions 1 and 2 motivate us to use the minimum among the $K$ available individual surprises for our intrinsic reward to reduce the gap between the actual target reward sum $\eta(\widetilde{\pi}^*)$ of the intrinsic reward-aided learned policy $\widetilde{\pi}^*$ and $\eta(\pi^*)$ of the true optimal policy $\pi^*$. Note that with the aid of intrinsic reward, we optimize $\widetilde{\eta}(\pi)$ in fact and this makes our policy (try to) approach $\widetilde{\pi}^*$ and the sample trajectory approach $(s,a) \sim \widetilde{\pi}^*$. So, with $\mathbb{E}_{(s,a)\sim\widetilde{\pi}^*}$ in the right-hand side of (11) replaced simply by the computable instantaneous sample-based value and $D_{KL}\left(P||P_{\phi^i}\right)|(s,a)$ replaced by the approximation (5), we propose using the minimum of $r^j_{t,int}(s_t, a_t, s_{t+1})$, $j = 1, \cdots, K$ as the single value of intrinsic reward from the $K$ candidates. That is, the agent selects the index $j^*$ as

$$j^* = \underset{1 \leq j \leq K}{\arg\min} \, r^j_{t,int}(s_t, a_t, s_{t+1}) \tag{12}$$

where $r^j_{t,int}$ is given by (7), and the intrinsic reward is determined as

$$r_{t,int}(s_t, a_t, s_{t+1}) = r^{j^*}_{t,int}(s_t, a_t, s_{t+1}). \tag{13}$$

## 2.3 IMPLEMENTATION

For the dynamics models $P_{\phi^1}, \cdots, P_{\phi^K}$, we adopted the fully-factorized Gaussian distributions (Achiam & Sastry, 2017; Houthooft et al., 2016). Then, $P_K$ in (4) becomes the class of $K$-modal Gaussian mixture distributions.

We first update the model ensemble $P_{\phi^1}, \cdots, P_{\phi^K}$ and the corresponding mixing coefficients $q_1, \ldots, q_K$. At the beginning, the parameters $\phi^1, \cdots, \phi^K$ are independently initialized, and $q_i$'s are set to $\frac{1}{K}$ for all $i = 1, \cdots, K$. At every batch period $l$, in order to jointly learn $\phi^i$ and $q_i$, we apply maximum-likelihood estimation with an $L_2$-norm regularizer with KL constraints (Achiam & Sastry, 2017; Williams & Rasmussen, 2006):

$$\underset{\phi^i\, q_i,\, 1\leq i \leq K}{\text{maximize}} \quad \mathbb{E}_{(s,a,s')} \log \left\{ \sum_{i=1}^{K} q_i P_{\phi^i}(s'|s,a) \right\} - \alpha \sum_{i=1}^{K} \|\phi^i\|^2 = \mathcal{L}_{\text{likelihood}} + \alpha\mathcal{L}_{\text{regularizer}}$$

$$\text{subject to} \quad \mathbb{E}_{(s,a)} \left[ D_{KL}(P_{\phi^i}||P_{\phi^i_{\text{old}}})(s,a) \right] \leq \kappa \quad (1 \leq i \leq K) \quad \text{and} \quad \sum_{i=1}^{K} q_i = 1 \tag{14}$$

where $\phi^i_{\text{old}}$ is the parameter of the $i$-th model before the update (14), $\alpha$ is the regularization coefficient, and $\kappa$ is a positive constant. To solve this optimization problem with respect to $\{\phi^i\}$, we apply the method based on second-order approximation (Schulman et al., 2015a). For the update of $\{q_i\}$, we apply the method proposed by Dempster et al. (1977) and set $q_i$ as follows:

$$q_i = \mathbb{E}_{(s,a,s')} \frac{q_i^{\text{old}} P_{\phi^i}(s'|s,a)}{\sum_{j=1}^{K} q_j^{\text{old}} P_{\phi^j}(s'|s,a)} \quad (1 \leq i \leq K) \tag{15}$$

where $q_i^{\text{old}}$ is the mixing coefficient of the $i$-th model before the update (15). For numerical stability, we use the "log-sum-exp" trick for computing (15) as well as $\mathcal{L}_{\text{likelihood}}$ and $\nabla_{\phi^i}\mathcal{L}_{\text{likelihood}}$. In addition, we apply simultaneous update of all $\phi^i$'s and $q_i$'s, which was found to perform better than one-by-one alternating update of the $K$ models for our problem.

Although the proposed intrinsic reward generation method can be combined with general RL algorithms, we here consider the PPO algorithm (Schulman et al., 2017), which is a popular on-policy algorithm and generates a batch of experiences of length $L$ with every current policy. Let $D$ be the batch of experiences for training the policy, i.e., $D = (s_t, a_t, r^{total}_t, s_{t+1}, \cdots, r^{total}_{t+L-2}, s_{t+L-1}, a_{t+L-1}, r^{total}_{t+L-1})$, where $a_t \sim \pi_{\theta_l}(\cdot|s_t)$, $s_{t+1} \sim P(\cdot|s_t, a_t)$, and $r^{total}_t$ is the total reward (16). Here, $\pi_{\theta_l}$ is the parameterized policy at the batch period $l$ corresponding to timestep $t, \cdots, t+L-1$ (the batch period index $l$ is now included in $\pi_{\theta_l}$ for clarity). The total reward at timestep $t$ for training the policy is given by

$$r^{total}_t(s_t, a_t, s_{t+1}) = r_t(s_t, a_t, s_{t+1}) + \beta\bar{r}_{t,int}(s_t, a_t, s_{t+1}) \tag{16}$$

---

**Algorithm 1** Sparse RL with Model Ensemble-Based Intrinsic Reward Based on PPO

---

1: $L$ : batch size for policy training, $L'$ : batch size for model training.
2: $L_{mini}$ : minibatch size for policy training, $L'_{mini}$ : minibatch size for model training.
3: $N$ : epoch size for policy training, $N'$ : epoch size for model training.
4: $MAX$ : the maximum index of batch period $l$, $K$ : the number of dynamics models.
5: Initialize the policy $\pi_{\theta_0}$, the $K$ transition probability models $P_{\phi_0^1}, \cdots, P_{\phi_0^K}$, and the correspond-
ing mixing coefficients $q_1, \cdots, q_K$.
6: Generate trajectories with $\pi_{\theta_0}$ and add them to the initially empty replay buffer $M$.
7: **for** Batch period $l = 0, \cdots, MAX - 1$ **do**
8:    Train $P_{\phi_l^1}, \cdots, P_{\phi_l^K}$ by performing gradient updates for (14), and update $q_1, \cdots, q_K$ by
performing iterations with (15). For this, we draw a batch $D'$ of size $L'$ randomly and
uniformly from $M$, and perform the updates with minibatches of size $L'_{mini}$ drawn from $D'$
for $N'$ epochs.
9:    **for** Timestep $t = lL, lL + 1, \cdots, lL + L - 1$ **do**
10:       Collect $s_t$ from the environment and $a_t$ with the policy $\pi_{\theta_l}$.
11:       Collect $s_{t+1}$ and the extrinsic reward $r_t$ from the environment and add $(s_t, a_t, s_{t+1})$ to $M$.
12:    **end for**
13:    Calculate the preliminary intrinsic reward $r_{t,int}$ in (13).
14:    Acquire the normalized intrinsic rewards of the current batch $D$ of size $L$ by using (17).
15:    Train $\pi_{\theta_l}$ by using PPO with the total rewards (16) and minibatch size $L_{mini}$ for $N$ epochs.
16: **end for**

---

where $r_t(s_t, a_t, s_{t+1})$ is the actual sparse extrinsic reward at timestep $t$ from the environment, $\bar{r}_{t,int}(s_t, a_t, s_{t+1})$ is the normalized intrinsic reward at timestep $t$, and $\beta > 0$ is the weighting factor. Note that the actually-used intrinsic reward $\bar{r}_{t,int}(s_t, a_t, s_{t+1})$ is obtained by applying normalization (Achiam & Sastry, 2017) to improve numerical stability as

$$\bar{r}_{t,int}(s_t, a_t, s_{t+1}) = \frac{r_{t,int}(s_t, a_t, s_{t+1})}{\max\left\{\frac{|\sum_{(s,a,s') \in D} r_{t,int}(s,a,s')|}{|D|}, 1\right\}} \tag{17}$$

where the unnormalized intrinsic reward $r_{t,int}(s_t, a_t, s_{t+1})$ is given by (13). Then, the policy $\pi_{\theta_l}$ can be updated at every batch period $l$ with $D$ by following the standard PPO procedure based on the total reward (16). Summarizing the above, we provide the pseudocode of our algorithm in Algorithm 1, which assumes PPO as the background algorithm. Note that the proposed intrinsic reward generation method can also be applied to other RL algorithms.

## 3 RESULTS

### 3.1 EXPERIMENTAL SETUP

In order to evaluate the performance, we considered sparse reward environments for continuous control. The considered tasks were five environments of Mujoco (Todorov et al., 2012), OpenAI Gym (Brockman et al., 2016): Ant, Hopper, HalfCheetah, Humanoid, and Walker2d. To implement sparse reward setting, we adopted the delay method (Oh et al., 2018). We first accumulate extrinsic rewards generated from the considered environments for every $\Delta$ timesteps or until the episode ends. Then, we provide the accumulated sum of rewards to the agent at the end of the $\Delta$ timesteps or at the end of the episode, and repeat this process. We set $\Delta = 40$ for our experiments.

All simulations were conducted over 10 fixed random seeds. The $y$-axis in each figure with the title "Average Return" represents the mean value of the extrinsic returns of the most recent 100 episodes averaged over the 10 random seeds. Each colored band in figure represents the interval of $\pm\sigma$ around the mean curve, where $\sigma$ is the standard deviation of the 10 instances of data from the 10 random seeds. (Please see Appendix B for detailed description of the overall hyperparameters for simulations.)

### 3.2 ABLATION STUDY

First, in order to validate the proposed approach in which the intrinsic reward is given by the minimum surprise over the ensemble, we investigated several methods of obtaining a single intrinsic reward value from the multiple preliminary reward values $r^1_{t,int}, \cdots, r^K_{t,int}$ in (7) from the $K$ models: the proposed minimum selection (12)-(13) and other possible methods such as the maximum selection, the average value taking method, and a pure 1-step surprise method with the mixture with the intrinsic reward defined as

$$r^{1-step,\ ens}_{t,int}(s_t, a_t, s_{t+1}) = \log \frac{\sum_{i=1}^{K} q_i P_{\phi^i_{l(t)}}(s_{t+1}|s_t, a_t)}{\sum_{i=1}^{K} q^{old}_i P_{\phi^i_{l(t)-1}}(s_{t+1}|s_t, a_t)}. \tag{18}$$

(18) results from the idea that we simply replace the unimodal model $P_\phi$ in (3) with the mixture model $P_K$ in (4).

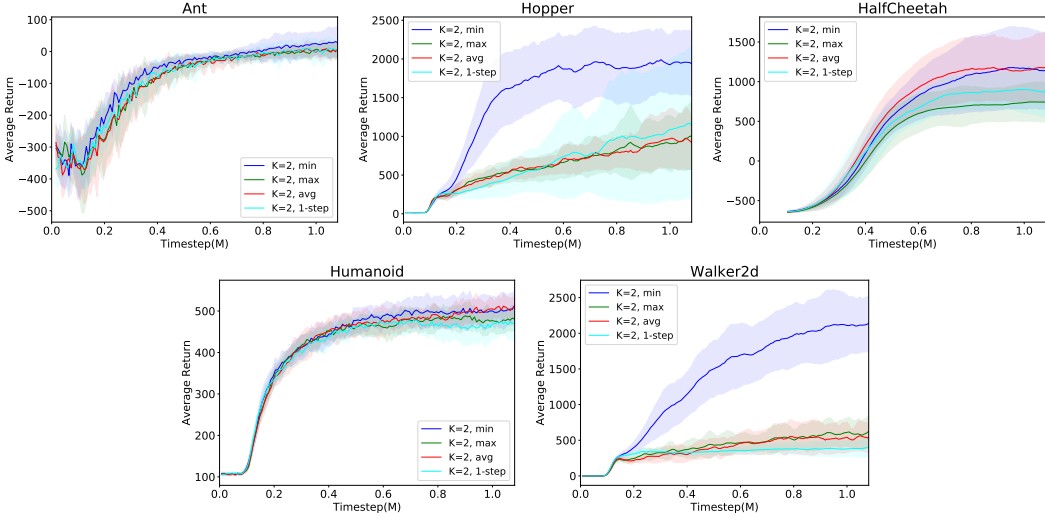

Figure 1: Impact of single intrinsic reward value extraction for $K = 2$: minimum selection (min), maximum selection (max), average (avg), and 1-step surprise with ensemble (1-step).

Fig. 1 shows the mean performance of the four single intrinsic reward value extraction methods for $K = 2$: the proposed minimum selection (12)-(13), the maximum selection, the average method, and the pure 1-step surprise with mixture (18). As inferred from Propositions 1 and 2, it is seen that the minimum selection yields the best performance in all the environments. (The average method yields similar performance in HalfCheetah and Humanoid.) Interestingly, the proposed approach motivated by Propositions 1 and 2 outperforms the simple mixture replacement in (18). With this validation, we use the minimum selection method (12)-(13) for all remaining studies.

Next, we investigated the impact of the model order $K$. Since we adopt Gaussian distributions for the dynamics models $P_{\phi^1}, \cdots, P_{\phi^K}$, the mixture $P_K$ in (4) is a Gaussian mixture for given state-action pair $(s, a)$. According to a recent result (Haarnoja et al., 2018), the model order of Gaussian mixture need not be too large to capture the underlying dynamics effectively in practice. Thus, we evaluated the performance for $K = 1, 2, 3, 4$.

Fig. 2 shows the mean performance as a function of $K$ for the considered sparse reward environments. It is observed that in general the performance improves as $K$ increases, and once the proper model order is reached, the performance does not improve further or degrades a bit due to more difficult model estimation for higher model orders, as expected from our intuition. From this result, we found that $K = 2$ seems reasonable for our model order, so we used $K = 2$ for all the five environments in the following performance comparison 3.3.

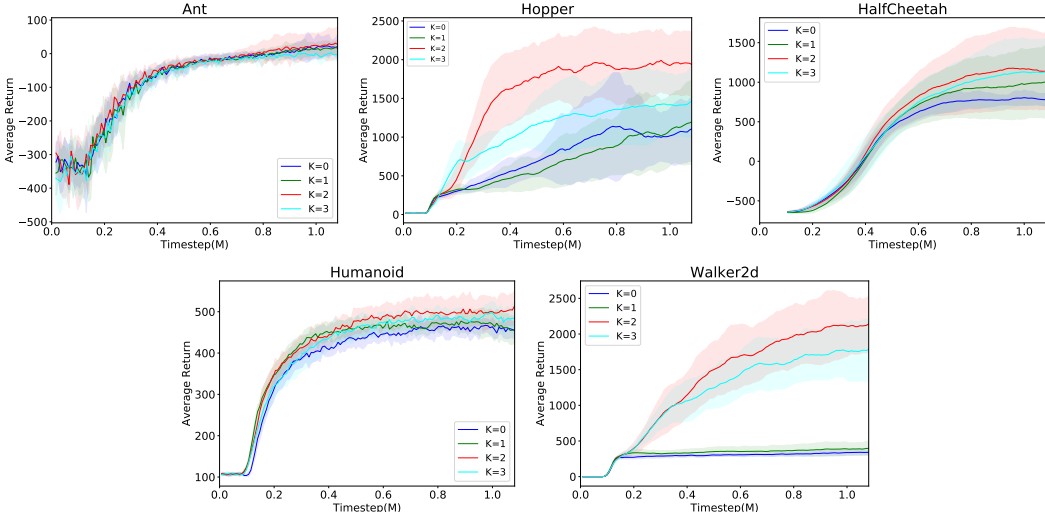

Figure 2: Mean performance for considered sparse reward environments as a function of $K$. $K = 0$ means PPO without intrinsic reward, and $K = 1$ means the single-model surprise method. ($K = 4$ yielded similar performance to that of $K = 3$, so we omitted the curve of $K = 4$ for simplicity)

### 3.3 PERFORMANCE COMPARISON

With the above verification, we compared the proposed method with existing intrinsic reward generation methods by using PPO as the background algorithm. We considered the existing intrinsic reward generation methods: curiosity (Pathak et al., 2017), hashing (Tang et al., 2017), information gain approximation (de Abril & Kanai, 2018), and single-model surprise (Achiam & Sastry, 2017). We also considered the method using intrinsic reward module (Zheng et al., 2018) among the most recent works introduced in Appendix C, which uses delayed sparse reward setup and provides an implementation code.

For fair comparison, we used PPO with the same neural network architecture and common hyperparameters, and applied the same normalization technique in (17) for all the considered intrinsic reward generation methods so that the performance difference results only from the intrinsic reward generation method. The weighting factor $\beta$ in (16) between the extrinsic reward and the intrinsic reward should be determined for all intrinsic reward generation methods. Since each of the considered methods yields different scale of the intrinsic reward, we used an optimized $\beta$ for each algorithm for each environment.

In the case of the single-model surprise method and the proposed method, the hyperparameters of the single-model surprise method are tuned to yield best performance and then the proposed method employed the same hyperparameters as the single-model surprise method. We also confirmed that the hyperparameters associated with the other four methods were well-tuned in the original papers (de Abril & Kanai, 2018; Pathak et al., 2017; Tang et al., 2017; Zheng et al., 2018), and we used the hyperparameters provided by these methods. (Please see Appendix B for detailed description of the hyperparameters for simulations.)

Fig. 3 shows the comparison results. It is seen that the proposed model-ensemble-based intrinsic reward generation method yields top-level performance. Note that the performance gain by the proposed method is significant in sparse Hopper and sparse Walker2d.

## 4 RELATED WORK

Various types of intrinsic motivation such as curiosity, information gain, and surprise have been investigated in cognitive science (Oudeyer & Kaplan, 2008), and intrinsically-motivated RL has been inspired from these studies. Houthooft et al. (2016) used the information gain on the dynamics model as additional reward based on the notion of curiosity. Pathak et al. (2017) defined an intrinsic reward

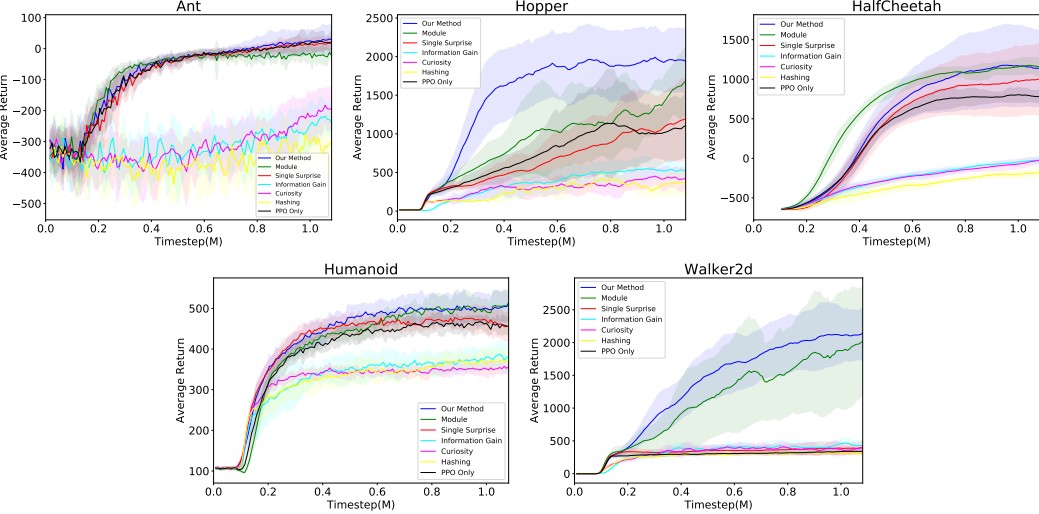

Figure 3: Performance comparison.

with the prediction error using a feature state space, and de Abril & Kanai (2018) enhanced Pathak et al. (2017)'s work with the idea of homeostasis in biology. The concept of surprise was exploited to yield intrinsic rewards (Achiam & Sastry, 2017).

In parallel with intrinsically motivated RL, researchers developed model-based approaches for learning itself, in which the agent uses the trained dynamics model and fictitious samples generated from the model for training. Nagabandi et al. (2017) suggested using the trained dynamics model to initialize the policy network at the beginning of model-free learning. Kurutach et al. (2018) proposed the policy optimization using trust-region method with a model ensemble, in which multiple prediction models for the next state for given pair of current state and action are constructed and trained by using actual samples, and the policy is trained by multiple fictitious sample trajectories from the multiple models. Our work differs from these works in that we use a model ensemble for the environment transition probability distribution and generates intrinsic reward based on this ensemble of dynamics models to enhance the performance of model-free RL with sparse reward. (Please see Appendix C for more related works.)

## 5  CONCLUSION

In this paper, we have proposed a new intrinsic reward generation method based on an ensemble of dynamics models for sparse-reward reinforcement learning. In the proposed method, the mixture of multiple dynamics models is used to better approximate the true unknown transition probability, and the intrinsic reward is designed as the minimum of the intrinsic reward computed from each dynamics model to the mixture to capture the most relevant surprise. The proposed intrinsic reward generation method was combined with PPO to construct a working algorithm. Ablation study has been performed to investigate the impact of the hyperparameters associated with the proposed ensemble-based intrinsic reward generation. Numerical results show that the proposed model-ensemble-based intrinsic reward generation method outperforms major existing intrinsic reward generation methods in the considered sparse environments.

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

## A   PROOFS

**Proposition 1.** *Let $\eta(\pi)$ be the actual expected discounted sum of extrinsic rewards defined in (8). Then, the following inequality holds:*

$$0 \underset{(a)}{\leq} \eta(\pi^*) - \eta(\widetilde{\pi}^*) \underset{(b)}{\leq} c\,\mathbb{E}_{(s,a)\sim\widetilde{\pi}^*}\left[D_{KL}\left(P||P'\right)|(s,a)\right] \tag{19}$$

*where $c$ is a positive constant.*

*Proof.* The inequality (a) is trivial from the definition of $\pi^*$, that is, $\pi^*$ is an optimal policy maximizing $\eta(\pi)$. The inequality (b) holds since

$$\eta(\pi^*) \leq \eta(\pi^*) + c\,\mathbb{E}_{(s,a)\sim\pi^*}\left[D_{KL}\left(P||P'\right)|(s,a)\right] \quad \text{since } c > 0, D_{KL}(\cdot||\cdot) \geq 0 \tag{20}$$

$$= \widetilde{\eta}(\pi^*) \quad \text{by definition of } \widetilde{\eta}(\cdot) \tag{21}$$

$$\leq \widetilde{\eta}(\widetilde{\pi}^*) \quad \text{by definition of } \widetilde{\pi}^* \tag{22}$$

$$= \eta(\widetilde{\pi}^*) + c\,\mathbb{E}_{(s,a)\sim\widetilde{\pi}^*}\left[D_{KL}\left(P||P'\right)|(s,a)\right] \quad \text{by definition of } \widetilde{\eta}(\cdot). \tag{23}$$

$\square$

**Proposition 2.** *Let $P_{\phi^i}(\cdot|s,a)$, $i = 1,\ldots,K$ be the ensemble of model distributions, and $P(\cdot|s,a)$ be an arbitrary true transition probability distribution. Then, the minimum of average KLD between $P(\cdot|s,a)$ and the mixture model $P' = \sum_i q_i P_{\phi^i}(\cdot|s,a)$ over the mixture weights $\{q_1, \cdots, q_K | q_i \geq 0, \sum_i q_i = 1\}$ is upper bounded by the minimum of average KLD between $P$ and $P_{\phi^i}$ over $\{i\}$: i.e.,*

$$\min_{q_1,\cdots,q_K} \mathbb{E}_{(s,a)\sim\widetilde{\pi}^*}\left[D_{KL}\left(P\,\middle\|\,\sum_i q_i P_{\phi^i}\right)\middle|(s,a)\right] \leq \min_i \mathbb{E}_{(s,a)\sim\widetilde{\pi}^*}\left[D_{KL}\left(P||P_{\phi^i}\right)|(s,a)\right]. \tag{24}$$

*Proof.*

$$\min_{\substack{q_1,\cdots,q_K \\ q_i \geq 0, \sum_i q_i = 1}} \left\{ \mathbb{E}_{(s,a)\sim\widetilde{\pi}^*}\left[D_{KL}\left(P\,\middle\|\,\sum_i q_i P_{\phi^i}\right)\middle|(s,a)\right] \right\} \tag{25}$$

$$\leq \min_{\substack{q_1,\cdots,q_K \\ q_i \geq 0, \sum_i q_i = 1}} \left\{ \mathbb{E}_{(s,a)\sim\widetilde{\pi}^*}\left[\sum_i q_i D_{KL}\left(P||P_{\phi^i}\right)\middle|(s,a)\right] \right\} \tag{26}$$

$$= \min_{\substack{q_1,\cdots,q_K \\ q_i \geq 0, \sum_i q_i = 1}} \left\{ \sum_i q_i \mathbb{E}_{(s,a)\sim\widetilde{\pi}^*}\left[D_{KL}\left(P||P_{\phi^i}\right)|(s,a)\right] \right\} \tag{27}$$

$$= \min_i \left\{ \mathbb{E}_{(s,a)\sim\widetilde{\pi}^*}\left[D_{KL}\left(P||P_{\phi^i}\right)|(s,a)\right] \right\}. \tag{28}$$

Here, (26) is valid due to the convexity of the KL divergence in terms of the second argument for a fixed first argument. (27) is valid due to the linearity of expectation. (28) is valid since the minimum in the right-hand side of (27) is achieved when we assign all the mass to $q_i$ that has the minimum value of $\mathbb{E}_{(s,a)\sim\widetilde{\pi}^*}\left[D_{KL}\left(P||P_{\phi^i}\right)|(s,a)\right]$. (Note that the optimal $\{q_i\}$ in (27) is not the same as the optimal $\{q_i\}$ achieving the minimum in (25).) $\square$

Note that each step in the proof is tight except (26) in which the convexity of the KL divergence in terms of the second argument is used. This part involves the function $f(x) = -\log x$ for $0 < x \leq 1$ since $D_{KL}(p_1||p_2) = \int p_1(y)\log\frac{p_1(y)}{p_2(y)}dy$, but the convexity of $f(x) = -\log x$ for $0 < x \leq 1$ is not so severe if $x$ is not so close to zero.

## B   Neural Network Architecture and Hyperparameters

For the actual implementation, the code implemented by Dhariwal et al. (2017) is used. The policy and dynamics models were designed by fully-connected neural networks all of which had two hidden layer of size (64, 64) (Dhariwal et al., 2017; Houthooft et al., 2016; Tang et al., 2017). The tanh activation function was used for all of the networks (Achiam & Sastry, 2017; Dhariwal et al., 2017). The means of the fully factorized Gaussian dynamics models were the outputs of our networks, and the variances were trainable variables which were initialized to 1 (Dhariwal et al., 2017). Other than the variances, all initialization is randomized so that each of dynamics models was set differently (Kurutach et al., 2018; Tavakoli et al., 2018). For the implementation of the policy model, our method and all the considered intrinsic reward generation method used the same code for the module method (Zheng et al., 2018).

Although a recent work (Achiam & Sastry, 2017) used TRPO (Schulman et al., 2015a) as the baseline learning engine, we used PPO (Schulman et al., 2017), one of the currently most popular algorithms for continuous action control, as our baseline algorithm. While the same basic hyperparameters as those in the previous work (Achiam & Sastry, 2017) were used, some hyperparameters were tuned for PPO. $\lambda$ for the GAE method (Schulman et al., 2015b) was fixed to 0.95, while the discounting factor was set to $\gamma = 0.99$. The batch size $L$ for the training of the policy was fixed to 2048. For the policy update using PPO, the minibatch size $L_{mini}$ was set to 64, the epoch number $N$ 10, the clipping constant 0.2, and the entropy coefficient 0.0. The maximum number of timesteps was 1M for all five environments, and the maximum index of batch period, $MAX$, is $\lfloor \frac{1M}{L} \rfloor = 488$. The learning late of Adam optimizer (Kingma & Ba, 2014) was fixed to 0.0003.

Each of the single-model surprise method, the hashing method and our proposed method requires a replay buffer. The size of the used replay buffer for all these three methods is 1.1M. Before the beginning of the iterations, $2048 \times B$ samples from real trajectories generated by the initial policy were added to the replay buffer. We set $B = 40$ for our experiments. For the methods not requiring a replay buffer, i.e., Curiosity, Information Gain, Module, and PPO Only, we ran $2048 \times B = 81920$ timesteps before measuring performance for fair comparison. (Therefore, every x-axis in Fig. 1, 2, and 3 shows the total timesteps of 1.08192M.)

For the dynamics model learning, we set the batch size $L' = 2048$, $L'_{mini} = 64$, and $N' = 4$. The optimization (14) was solved based on second-order approximation (Schulman et al., 2015a). When $K = 1$, the optimization (14) reduces to the model learning problem in Achiam & Sastry (2017). In Achiam & Sastry (2017), the constraint constant $\kappa$ in the second-order optimization was well-tuned as 0.001. So, we used this value of $\kappa$ not only to the case of $K = 1$ but also to the case of $K \geq 2$. We further tuned the value of $\alpha$ in (14) for each environment, and we set $\alpha = 0.01$. For the information gain method, we need another hyperparameter $h$ which is the weight to balance the original intrinsic reward and the homeostatic regulation term (de Abril & Kanai, 2018). We tuned this hyperparameter for each environment and the used value of $h$ is shown in Table 1.

Table 1 summarizes the weighting factor $\beta$ as well as the hyperparameter $h$ in information gain method. Here, we used the optimized weighting factor $\beta$ in (16) for each algorithm for each environment. As aforementioned, the major hyperparameters for the proposed model-ensemble-based method are the same as those used for the single-model surprise method.

|  | Ant | Hopper | HalfCheetah | Humanoid | Walker2d |
|---|---|---|---|---|---|
| Curiosity | $\beta = 0.01$ | $\beta = 0.01$ | $\beta = 0.0001$ | $\beta = 0.1$ | $\beta = 0.03$ |
| Hashing | $\beta = 0.0001$ | $\beta = 0.01$ | $\beta = 0.00001$ | $\beta = 0.1$ | $\beta = 0.003$ |
| Information Gain | $\beta = 0.01, h = 4$ | $\beta = 0.1, h = 4$ | $\beta = 0.0001, h = 4$ | $\beta = 0.01, h = 2$ | $\beta = 0.03, h = 2$ |
| Single Surprise | $\beta = 0.00001$ | $\beta = 0.05$ | $\beta = 0.0001$ | $\beta = 0.3$ | $\beta = 0.05$ |

Table 1: Used hyperparameter values.

For the intrinsic reward module method, we checked that the provided source code in github reproduced results in Zheng et al. (2018), as shown in Fig. 4. 'Module 0.01' represents the module method with training using the sum of intrinsic reward and scaled extrinsic reward with scaling factor 0.01. 'Module 0' represents training using intrinsic reward only (no addition of extrinsic reward). Both methods are introduced in Zheng et al. (2018), and we checked reproducibility when $B = 0$, i.e., we

ran $2048 \times B = 0$ timesteps before measuring performance. We observed that our used code yielded the same results as those in Zheng et al. (2018).

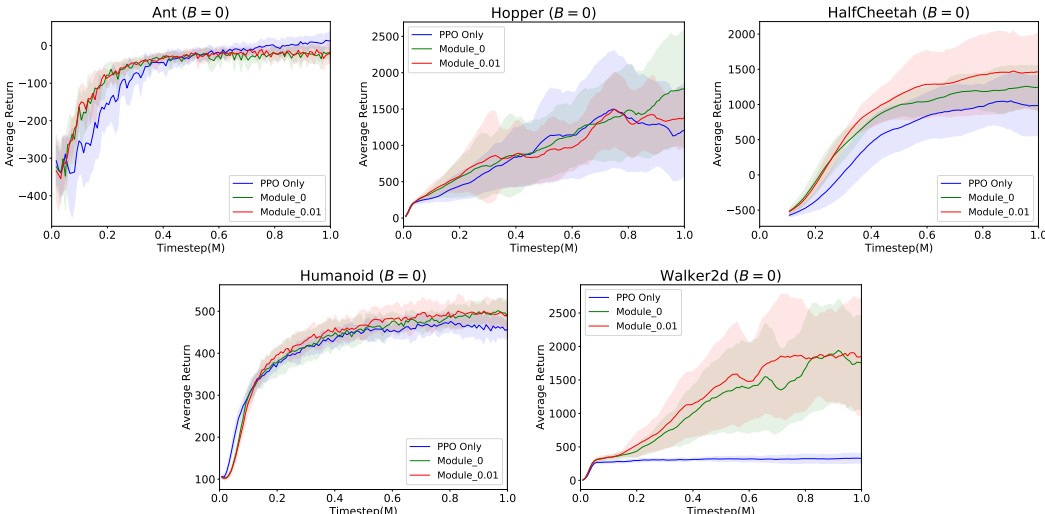

Figure 4: Reproduced mean performance of the module method over 10 random seeds with $\Delta = 40$ when $B = 0$.

Thus, we used this code for the module method with only one change that we ran $2048 \times B$ timesteps with $B = 40$ before measuring performance for fair comparison. (Since the range of intrinsic reward from the module method is $[-1, 1]$, intrinsic reward normalization in (17) is not needed.) For the module method in performance comparison 3.3, we selected a better method between 'Module 0' and 'Module 0.01', assuming $B = 40$. 'Module 0' performed better than 'Module 0.01' in Hopper and Walker2d, and 'Module 0.01' performed better than 'Module 0' in the other three environments.

## C    MORE RELATED WORK

Recent advanced exploration methods can be classified mainly into two categories. One is to generate intrinsic reward explicitly and to train the agent with the total reward which is the sum of the extrinsic reward and the adequately scaled intrinsic reward. The other is indirect methods which do not explicitly generate intrinsic reward. Our work belongs to the first category. There exist many exploration techniques on image spaces (Bellemare et al., 2016; Burda et al., 2018a;b; Savinov et al., 2018) but these works are not directly related to our work here.

### 1. Explicit Intrinsic Reward Generation

Andrychowicz et al. (2017) suggested a new intrinsic reward for sparse and binary extrinsic reward environments, based on sampling additional states from the replay buffer and setting those data as new goals. In their work the policy was based on the input of both state and goal. In our work, on the other hand, the concept of goal is not necessary. A recent work by Zheng et al. (2018) used a delayed reward environment to propose training the module to generate intrinsic reward apart from training the usual policy. This delayed reward environment for sparse reward setting is different from the previous sparse reward environment based on thresholding (Houthooft et al., 2016), i.e., the agent get non-zero reward when the agent achieves a certain physical quantity (such as the distance from the origin) larger than the predefined threshold. Recently, Freirich et al. (2018) proposed generating intrinsic reward by applying a generative model with the Wasserstein-1 distance. With the concept of state-action embedding, Kim et al. (2018) adopted the Jensen-Shannon divergence (JSD) (Hjelm et al., 2018) to construct a new variational lower bound of the corresponding mutual information, guaranteeing numerical stability. Our work differs from these two recent works in that we used a model ensemble to generate intrinsic reward.

### 2. Exploration without Intrinsic Reward Generation

Recent indirect methods can further be classified mainly into two groups: (i) revising the original objective function to stimulate exploration, which exploits intrinsic motivation implicitly, and (ii) perturbing the parameter space of policy.

In the first group, Oh et al. (2018) proposed that exploration can be stimulated by exploiting novel state-action pairs from the past, and used sparse reward environments by delaying extrinsic rewards. Hong et al. (2018) revised the original objective function for training by considering maximization of the divergence between the current policy and recent policies, with an adaptive scaling technique. The concept of dropout was applied to the PPO algorithm to encourage the stochastic behavior of the agent episode-wisely (Xie et al., 2018). Convex combination of the target policy and any given policy is considered as a new exploratory policy (Shani et al., 2018), which corresponds to solving a surrogate Markov Decision Process, generalizing usual exploration methods such as $\epsilon$-greedy or Gaussian noise.

In the second group, Colas et al. (2018) proposed a goal-based exploration method for continuous control, which alternates generating parameter-outcome pair and perturbing certain parameters based on randomly drawn goal from the outcome space. Recently, inspired by Chua et al. (2018), Shyam et al. (2018) considered pure exploration MDP without any extrinsic reward with the notion of utility, where utility is based on JSD and the Jensen-Rényi divergence (Rényi et al., 1961). In this work, they consider a number of models for transition function but they used this to compute utility based on average entropy of the multiple models. Our work uses the minimum of surprise from the multiple dynamics models under the existence of explicit reward whether it is extrinsic or intrinsic.

