# OpenReview forum: "Model Ensemble-Based Intrinsic Reward for Sparse Reward Reinforcement Learning"
_ICLR.cc/2020/Conference — Reject_

### Official Review · AnonReviewer1 · 2019-10-21
**Official Blind Review #1**

**Rating:** 3

**Review:**

Summary

This paper proposes a model-based method for intrinsic rewards based on probabilistic neural network ensembles. For a particular ensemble element, an intrinsic reward is defined as a log-term that measures the deviation in prediction between a Gaussian mixture over all ensemble elements and the particular ensemble element (at the previous update period). The intrinsic reward signal applied in practice is the minimum of the aforementioned quantity w.r.t. all ensemble elements. The authors provide some theoretical motivation for their approach and validate their method in sparse-reward continuous robotics domains (Mujoco), using PPO as reference algorithm. Experiments are averaged over 10 seeds and compared against various other intrinsic reward baselines including PPO without intrinsic motivation. In these experiments, the method provided by the authors seems to dominate over competing approaches.

Quality

I find the quality low. First, while I appreciate that the authors try to provide some theoretical motivation for their method, there is quite a gap from theory to practice. The theory assumes that the world model is known and considers a stationary reward signal. In practice, the world model is approximated with the current network ensemble, a quantity that is changing over time yielding a non-stationary signal. Second, the experiments are conducted in a sparse-reward modification of Mujoco-type environments that are non-sparse by construction. Sparsity is introduced by accumulating instantaneous rewards over some time window before "releasing" them. This way of sparsifying rewards yields weird partial observability issues, e.g. the same state-action pair observed at the right moment in time yields significant reward whereas at the wrong moment in time yields no reward at all. Additionally, it is always guaranteed that there will be a reward signal at a certain frequency. There are probably better environments for studying the approach, like Atari for example. I do understand that the authors cite Oh et al. 2018 who apply the same technique of sparsification, but Oh et al. also conduct additional experiments in ALE.

Clarity

The paper is clearly written and easy to follow. On a side note, it could be stated more clearly that the presented approach is not model-based because no forward prediction is required for constructing intrinsic rewards (merely probability values are queried for observed transitions). There is one question I have though regarding the experiments where different intrinsic reward approaches are compared against each other. The paper states that all approaches normalize intrinsic rewards according to Equation (17)---why do all of them then need a different weighting factor \beta as stated in the second paragraph of Section 3.3? Furthermore, since \beta is fine-tuned for each intrinsic reward approach and each environment, can the authors please elaborate in detail how exactly this is accomplished (to ensure correct interpretability of the results)?

Originality

The originality is low. As stated by the authors, the proposed method is an incremental extension of Achiam and Sastry 2017 who proposed a similar method for non-ensemble methods.

Significance

The significance is low as well. The method's improvement over other intrinsic reward approaches is minor (the environments chosen by the authors are also not ideal). I feel the significance is reduced further by the fact that there are other model-based approaches that use models of the proposed kind for increasing sample-efficiency and performance considerably in non-sparse environments (e.g. Kurutach et al.2018).

Update

After reading the other reviews and the rebuttal, the concerns I have raised remain. I still feel this paper shouldn't be accepted to ICLR. However, given the extensive experimental analysis conducted by the authors, I feel a score of 1 from my side was too harsh. I therefore increase to 3.

**Experience Assessment:**

I have published one or two papers in this area.

**Review Assessment: Checking Correctness Of Derivations And Theory:**

I assessed the sensibility of the derivations and theory.

**Review Assessment: Checking Correctness Of Experiments:**

I assessed the sensibility of the experiments.

**Review Assessment: Thoroughness In Paper Reading:**

I read the paper at least twice and used my best judgement in assessing the paper.

---

> ### Author Response · Authors · 2019-11-14
> **Response to Reviewer #1**
>
> We would like to thank you for your valuable comments. Although our work focused on designing an intrinsic reward on continuous state spaces (as mentioned at the beginning of the Preliminaries subsection), we fully accept your thoughtful opinion. Therefore, we leave our application for image-based countable state spaces as our anticipating future work.
>
> We admit the comment that we approximated the true model to the ensemble using the inequality (5), and need more experiments using other environments. We would like to estimate the likelihood ratio and conducting experiments of other environments as our other future work.
>
> For a fair comparison, we followed the same normalization technique (Eq. (17)) as in Achiam and Sastry (2017). As we replied to Reviewer 3, the mean curves of accumulated intrinsic reward for our approach converge to 0 rapidly, so the denominator of the right-hand-side of Eq. (17) is 1. Therefore, this normalization does not guarantee that each of the considered intrinsic reward generation methods produces the same scale of intrinsic reward. Therefore, we additionally need fine-tuning of \beta for each method and each environment.
>
> We swept our hyperparameter \beta according to log scale (Zheng, Oh, and Singh, "On Learning Intrinsic Rewards for Policy Gradient Methods", 2018): {1.0, 0.5, 0.3, 0.2, 0.1, 0.05, 0.03, 0.02, 0.01, … }. For Ant and HalfCheetah, we observed that smaller \beta should be tested, so we considered \beta to the scale of 10^(-7).
>
> Our work differs from the model-based approach such as Kurutach et al.’s (2018) in that while this work focused on non-sparse reward environments, we use a model ensemble for generating intrinsic rewards to enhance the performance of model-free RL in sparse reward setting.

---

### Official Review · AnonReviewer3 · 2019-10-21
**Official Blind Review #3**

**Rating:** 6

**Review:**

This paper presents an approach for using an ensemble of learning dynamics models to generate an intrinsic reward for reinforcement learning in sparse reward environments. The paper's results demonstrate that this approach out-performs other benchmark approaches across a set of continuous control tasks.

I'm curious about the motivation for taking the min surprise across the ensemble. Eqs 8-11 seem to motivate that this will make a tighter gap between the expected cumulative returns of the optimal policy and the policy with additional intrinsic rewards. This implies that this will converge to the same policy in the end, but I'm not sure that it implies that this is a good exploration strategy, an intrinsic reward of 0 would provide an even tighter bound. When you used the max or average return, was the problem that there was still too much intrinsic reward for the agent to converge? Or that it wasn't exploring enough or in the right places?

For the experiments, it would be very interesting to see the intrinsic rewards accumulated over time for each approach. That plot could be very enlightening as to what is going on.

It would also be great to see the results on these tasks when they're not modified to be sparse reward. Is your method a big hindrance in that case? Or does it still help?

Your intro claims that typical model-free RL is about the circumstance where the agent receives non-zero reward for every time step. This is not true, many model-free RL papers look at tasks that have sparse reward on some or many steps.  I would agree that in many continuous control problems, shaping rewards are added to ease the exploration problem.

For Figure 3, how does the model-ensemble TRPO (Kurutach et al) fit in? Is that algorithm represented by one of the curves?

Here's one more related work deriving intrinsic rewards from an ensemble of trained dynamics models:
http://www.sciencedirect.com/science/article/pii/S0004370215000764

**Experience Assessment:**

I have published one or two papers in this area.

**Review Assessment: Checking Correctness Of Derivations And Theory:**

I assessed the sensibility of the derivations and theory.

**Review Assessment: Checking Correctness Of Experiments:**

I carefully checked the experiments.

**Review Assessment: Thoroughness In Paper Reading:**

I read the paper thoroughly.

---

> ### Author Response · Authors · 2019-11-14
> **Response to Reviewer #3**
>
> We would like to thank you for your detailed comments.
> Motivations for taking the min surprise across the ensemble are our Proposition 1 and 2: the minimum selection gives tightest upper bound on the expected KL divergence between true P and the ensemble P_K within the class of linear combinations of the individual surprise values. The maximum and average method cannot give this tightest upper bound.
>
> An intrinsic reward of 0 would provide the theoretically tightest bound in Proposition 1 (c=0), but our setting is that extrinsic reward is zero at the beginning of our training, so we cannot train our policy in this sparse-reward setting if we do not give any intrinsic reward.
>
> The mean curves of accumulated intrinsic reward for our approach converge to 0 at the end. This observation is valid since as our ensemble models are trained, we get smaller expected KL divergence between true P and the ensemble P_K and its corresponding intrinsic reward.
>
> Whether a certain type of intrinsic reward is helpful for non-sparse reward environments depends on the intrinsic reward coefficient tuning as well as the type of certain intrinsic reward. Too big hyperparameter hindered the performance, and some fine-tuned hyperparameters can improve the performance in a non-sparse reward setting, as shown in the recent work (Leibfried, Pascual-Diaz, and Grau-Moya, "A unified Bellman optimality principle combining reward maximization and empowerment", 2019).
>
> We did not use model-ensemble TRPO (Kurutach et al., 2018) in Figure 3 since this paper has a model-based setting while our paper proposes an intrinsic reward for model-free reinforcement learning. In other words, as Reviewer 1 pointed out, we do not require any forward prediction for constructing auxiliary reward as well as training policy.
>
> We read the recommended paper (Hester and Stone, 2015) and we would like to appreciate your kind suggestion. We liked the idea of using two kinds of different intrinsic rewards as well as training our model with a random-forest based method. We expect this paper can be applied to our extended future work!

---

> > ### Comment · AnonReviewer3 · 2019-11-14
> > **Response**
> >
> > Thanks for your response. I wonder what your thoughts are on whether getting a tight upper bound on the KL divergence between the true P and the ensemble P_K is the right goal? Because then you state that you are required to have some intrinsic reward to train the policy. It's not clear to me that this theoretical result actually indicates what you should do in practice. Yes, you don't want to differ in the limit, but during exploration you do need these to be different.
> >
> > The accumulated intrinsic rewards approach 0 meaning that initially you're getting positive rewards and then you start getting negative to bring the cumulative sum back to 0? Such a plot of the intrinsic rewards I think would be valuable in understanding the algorithm.

---

> > > ### Author Response · Authors · 2019-11-15
> > > **Response to Reviewer 3**
> > >
> > > We would like to thank you for your other comment. As you mentioned, we need intrinsic rewards during exploration, but we observed that intrinsic rewards of large values did not necessarily help training policy when extrinsic reward was sparse. In other words, the difference between P and P_K should be "different but not too much." Minimum selection can prevent overestimation of surprise as well as reduce the gap in Proposition 1. Preventing overestimation using minimum selection was in fact shown to be valid in previous works such as Twin delayed DDPG, or TD3* (although the goal and basic setting of our paper are different from TD3).
> > >
> > > The accumulated intrinsic rewards approach 0 meant that we initially got a lot of positive intrinsic rewards and a relatively small amount of negative intrinsic rewards. Then we started getting positive and negative intrinsic rewards of small absolute values to bring the cumulative sum of intrinsic reward close to 0. As you mentioned, we needed intrinsic reward for exploration, but then we got nonzero extrinsic reward after exploration and the role of intrinsic reward should be reduced, which was accomplished by training our model P_K close to true P.
> > >
> > > * Fujimoto et al., "addressing function approximation error in actor-critic methods", 2018.

---

### Official Review · AnonReviewer2 · 2019-10-23
**Official Blind Review #2**

**Rating:** 3

**Review:**


This paper proposes an auxiliary reward for model-based reinforcement learning. The proposed method uses ensembles to build a better model of environment dynamics and suggests some rules to optimize the new ensemble-based dynamics and to estimate the intrinsic reward.

I am torn on this paper. I like the derivation of the method and the ideas behind it. I think it is an interesting direction of research. However, the experiments are limited to one domain and the paper needs proofreading. I will vote "weak accept" for this paper, as I think it is incremental and the experiments are too limited.

As I said above, the paper could use some proofreading. Some sections (like pages 1-2) are well written, but others are full of grammatical mistakes. There is also a lot of redundant information.

It is often stated that the ensemble model has better capacity than the single model. Some experimental proof of this better modelling capacity could help convince a reader that the ensemble is indeed beneficial and warranted (e.g., show that P_K is better than the single model P).

Some evaluation or discussion on the computational costs of the method would be beneficial. I assume the ensemble-based method is more computationally intensive. Would it perform better than single surprise if they were compared according to wall clock time?


Examples of minor issues:

- Page 3, after equation 7, sentence beginning with "In addition to the advantage that the mixture model (4) has the increased model order for modelling" is confusing, contains redundant elements, and is not bringing useful information. It should be revised
- page 4, in paragraph after eq. 11: the following sentence is grammatically incorrect, please revise: "Propositions 1 and 2 suggest a way to intrinsic reward design under the mixture model for tight gap between η(π ̃∗) and η(π∗)"
- in the same sentence, revise: "be close to the true η(π∗) of the true optimal policy π∗, as desired.",
- page 6 text above Figure 1: "single-modal" should be unimodal


POST REBUTTAL

Thanks for writing your rebuttal. I have read it, as well as the other reviews. I think reviewer 1 touches on some important points, especially regarding the engineered sparse rewards. It seems the method is not properly justified given the environments used for its evaluation. Based on this, and the fact that the method is rather incremental, I would like to change my score to a weak reject. The method should be evaluated in a setting­ with truly sparse rewards.

**Experience Assessment:**

I have read many papers in this area.

**Review Assessment: Checking Correctness Of Derivations And Theory:**

I assessed the sensibility of the derivations and theory.

**Review Assessment: Checking Correctness Of Experiments:**

I carefully checked the experiments.

**Review Assessment: Thoroughness In Paper Reading:**

I read the paper thoroughly.

---

> ### Author Response · Authors · 2019-11-14
> **Response to Reviewer #2**
>
> We would like to thank you for your kind comments. Although our work focused on designing an intrinsic reward on continuous state spaces (as mentioned at the beginning of the Preliminaries subsection), we fully accept your thoughtful opinion. Therefore, we leave our application for image-based countable state spaces as our anticipating future work.
>
> The fact that the K ensemble model has better capacity than the single model is due to the principle of EM algorithm (Dempster, Laird, and Rubin, 1977). EM algorithm finds the best mixing coefficients q_1, … , q_K  of the underlying dynamics and this algorithm can cover M modes of the dynamics if M <= K. Therefore, the more models our ensemble has, the better capacity it maintains. However, this ensemble has K times more computational costs than the single model case. We leave reducing computation cost problems as our other future work.
>
> One thing we need to clarify is that, as Reviewer 1 pointed out, our paper proposes an intrinsic reward for model-free reinforcement learning, since no forward prediction is required for constructing auxiliary reward.
>
> We proofread our paper according to your meticulous comments and uploaded a new version of the paper.

---

### Decision · Program_Chairs · 2019-12-19

**Decision:**

Reject

**Comment:**

This paper considers the challenge of sparse reward reinforcement learning through intrinsic reward generation based on the deviation in predictions of an ensemble of dynamics models. This is combined with PPO and evaluated in some Mujoco domains.

The main issue here was with the way the sparse rewards were provided in the experiments, which was artificial and could lead to a number of problems with the reward structure and partial observability. The work was also considered incremental in its novelty. These concerns were not adequately rebutted, and so as it stands this paper should be rejected.